# SCALABLE AND EFFECTIVE IMPLICIT GRAPH NEURAL NETWORKS ON LARGE GRAPHS

**Juncheng Liu   Bryan Hooi   Kenji Kawaguchi   Yiwei Wang**[†]   **Chaosheng Dong**[‡]   **Xiaokui Xiao**
National University of Singapore    [†]University of California, Los Angeles    [‡]Amazon.com Inc.
`juncheng.liu@u.nus.edu`

## ABSTRACT

Graph Neural Networks (GNNs) have become the de facto standard for modeling graph-structured data in various applications. Among them, implicit GNNs have shown a superior ability to effectively capture long-range dependencies in underlying graphs. However, implicit GNNs tend to be computationally expensive and have high memory usage, due to 1) their use of full-batch training; and 2) they require a large number of iterations to solve a fixed-point equation. These compromise the scalability and efficiency of implicit GNNs especially on large graphs. In this paper, we aim to answer the question: *how can we efficiently train implicit GNNs to provide effective predictions on large graphs?* We propose a new scalable and effective implicit GNN (SEIGNN) with a mini-batch training method and a stochastic solver, which can be trained efficiently on large graphs. Specifically, SEIGNN can more effectively incorporate global and long-range information by introducing coarse-level nodes in the mini-batch training method. It also achieves reduced training time by obtaining unbiased approximate solutions with fewer iterations in the proposed solver. Comprehensive experiments on various large graphs demonstrate that SEIGNN outperforms baselines and achieves higher accuracy with less training time compared with existing implicit GNNs.

## 1   INTRODUCTION

Recently, Graph Neural Networks (GNNs) have been widely used for modeling graph-structured data in the real world and achieved great success in numerous applications (Wu et al., 2020) including computer vision (Shi and Rajkumar, 2020), recommendation systems (Zhang et al., 2020), and drug discovery (Wan et al., 2019). In general, to utilize both graph topology and node attributes for generating meaningful node representations, modern GNNs iteratively aggregate representations of neighbors of each node with its own representation to update representations, which is termed as the "message passing" mechanism (Gilmer et al., 2017).

Despite the success achieved by these GNNs on different tasks, they lack the ability to capture long-range dependencies in graphs. The reason is that traditional GNN models can only capture information up to $T$-hops away with $T$ layers. $T$ cannot be large because a large $T$ causes the over-smoothing problem (Li et al., 2018). To mitigate this limitation of traditional GNNs, another type of GNNs, called implicit GNNs, has been proposed to capture long-range dependencies (Gu et al., 2020; Liu et al., 2022; Li et al., 2023). Implicit GNNs define an implicit layer using a fixed-point equation for aggregation and generate the equilibrium as node representations by solving the fixed-point equation. These implicit GNNs have superiority in capturing long-range information as they can be treated as a GNN with many layers defined implicitly to aggregate the information from distant nodes in the forward pass. Meanwhile, they enjoy constant memory complexity through implicit differentiation when computing gradients in the backward pass (Bai et al., 2019).

In spite of the advantages of implicit GNNs in capturing long-range information, they suffer from scalability issues on large graphs. **The first limitation** is is their reliance on full-batch training, which involves iteratively aggregating an entire graph to solve the fixed-point equation. This approach incurs significant computational costs and it is sometimes not even feasible on large graphs since massive memory may be required to load a whole graph into GPU during training. For example, MGNNI runs out of GPU memory on the ogbn-products dataset as shown in Table 2.

Although there are several sampling-based methods proposed to enable mini-batch training for traditional GNN models (Hamilton et al., 2017; Zeng et al., 2021; Chiang et al., 2019), it is infeasible to directly apply these methods to implicit GNNs without sacrificing their effectiveness [1]. The reason is that these methods split an entire graph into several subgraphs as mini-batches, which prohibits information propagation between different mini-batches. Therefore, directly employing existing mini-batch methods may be harmful to the ability of implicit GNNs to capture long-range information. Besides the limitation of using full-batch training, **the second limitation** is that implicit GNNs are computationally expensive to train, as they usually require a large number of iterations to iteratively solve a fixed-point equation. This issue can be exacerbated on large graphs.

Motivated by the aforementioned limitations of previous implicit GNNs, in this paper, we aim to answer the question: *how to efficiently train implicit GNNs to provide effective predictions on large graphs?* To achieve this, we propose a scalable and effective implicit GNN (SEIGNN) with a mini-batch training method. Specifically, following previous sampling-based methods, our designed mini-batch training method also samples subgraphs, but adds coarse-level nodes representing different graph partitions, while new edges are included considering both coarse nodes and original nodes. In this way, our mini-batch training method avoids full-batch training and encourages information propagation between nodes within different mini-batches, which cannot be achieved by previous implicit GNNs or by directly applying existing mini-batch methods to implicit GNNs. Therefore, with this mini-batch training, SEIGNN can scale up to large graphs without sacrificing the ability to capture global or long-range information.

Moreover, to reduce the extensive training time of previous implicit GNNs caused by a large number of iterations for obtaining equilibrium, in SEIGNN, we also propose a new stochastic unbiased solver that can solve the fixed-point equation with fewer iterations to obtain approximated equilibrium.

**Our contributions**  We summarize the contributions of this work as follows:

- We first point out the scalability and efficiency limitations of previous implicit GNNs, which are caused by full-batch training and their large number of iterations used for getting equilibrium.
- To mitigate the limitations, we propose SEIGNN, a scalable and effective implicit GNN, which incorporates our designed mini-batch training methods with coarse nodes to capture global information; and a new stochastic solver that achieves unbiased estimates of equilibriums. Through these, SEIGNN can be efficiently trained on large graphs while maintaining the ability to capture long-range information.
- Comprehensive experiments on 6 datasets show that SEIGNN achieves better accuracies with less training time on large graphs. Additionally, the detailed ablation studies further demonstrate the effectiveness of our methods.

## 2 RELATED WORK

### 2.1 GRAPH NEURAL NETWORKS

GNNs have been widely used for modeling graph data in different tasks. Modern GNNs (Kipf and Welling, 2016; Hamilton et al., 2017) usually follow the "message passing" mechanism where information is aggregated from the neighbor nodes for each node. Different GNN models have been proposed to utilize skip connection (Xu et al., 2018; Chen et al., 2020), attention mechanism (Veličković et al., 2018), and simplified activation (Wu et al., 2019). However, these models usually only involve finite aggregation layers due to the over-smoothing problem (Li et al., 2018), which makes them hardly capture long-range dependencies on graphs.

### 2.2 IMPLICIT MODELS AND IMPLICIT GRAPH NEURAL NETWORKS

Implicit models / deep equilibrium models (Bai et al., 2019) generally define an equilibrium equation as implicit hidden layers to generate outputs by solving the equation. These models have attracted much attention recently as they can avoid storing hidden states and achieve constant memory consumption by using implicit differentiation. For example, Bai et al. (2019) and Bai et al. (2020)

---

[1]Table 5 empirically shows that directly using these methods with implicit GNNs performs poorly.

propose the deep equilibrium model (DEQ) and its multiscale variant demonstrating the ability of implicit models in image and text related tasks. Some theoretical works (Kawaguchi, 2021) and (Geng et al., 2021) also explore the convergence analysis and provide a gradient estimate for implicit models to avoid exact gradient computation, respectively.

Inspired by implicit models, several implicit GNN models have been proposed, such as IGNN (Gu et al., 2020), EIGNN (Liu et al., 2021), MGNNI (Liu et al., 2022), CGS (Park et al., 2022). Since these models iteratively aggregate information from neighbors to obtain the fixed-point solution of the equilibrium equation, they can capture long-range information on graphs. However, as these implicit GNNs use full-batch training that iteratively aggregates a full graph to get outputs, they cost a lot of training time and need massive memory for storing a whole graph and the representation of each node in GPU memory. To reduce the computation complexities, a recent work USP (Li et al., 2023) proposes to use mini-batch training as aggregating information on randomly sampled subgraphs with proposed proximal solvers. However, randomly sampling subgraphs as mini-batch might affect the ability of implicit GNNs to capture long-range information.

## 2.3 GRAPH NEURAL NETWORKS FOR LARGE GRAPHS

To avoid huge computation costs incurred in training GNNs on large graphs, several mini-batch training methods for traditional GNNs have been proposed to scale up GNNs. Cluster-GCN (Chiang et al., 2019), GraphSAINT (Zeng et al., 2020), and Shadow-GNN (Zeng et al., 2021) sample subgraphs as minibatches and train GNNs within different subgraphs to reduce the computation cost. Specifically, Cluster-GCN relies on a graph clustering method to generate subgraphs while Shadow-GNN uses Personalized-PageRank scores to select important nodes to form a subgraph for each target node. GraphSAGE (Hamilton et al., 2017) proposes to use a sampled neighborhood of a node for message aggregation to efficiently generate node representations.

## 3 PRELIMINARIES

An undirected graph can be represented as $\mathcal{G} = (\mathcal{V}, \mathcal{E})$ which consists of the node set $\mathcal{V}$ with $n$ nodes and the edge set $\mathcal{E}$. Each node $v$ has a length-$d$ feature $x_v$. The adjacency matrix $A \in \mathbb{R}^{n \times n}$ and the node feature matrix $X \in \mathbb{R}^{d \times n}$ are taken as the input for graph neural networks. Considering unweighted adjacency matrix $A$, if node $i$ and $j$ are connected, $A_{i,j} = 1$, otherwise $A_{i,j} = 0$.

**Traditional GNNs and Implicit GNNs** Traditional GNNs (Kipf and Welling, 2016; Chen et al., 2020; Hamilton et al., 2017) have a learnable aggregation process that iteratively propagates information from each node to its neighbor nodes. For each layer $l$, the aggregation step can be defined as follows:

$$Z^{(l+1)} = \phi(W^{(l)} Z^{(l)} S + \Omega^{(l)} X), \tag{1}$$

where $Z^{(l)}$ is the hidden states in layer $l$, $S$ is the normalized adjacency matrix; $W^{(l)}$ and $\Omega^{(l)}$ are trainable weight matrices.

Similar to traditional GNNs, implicit GNNs (Gu et al., 2020; Liu et al., 2021; 2022; Park et al., 2022) also have an aggregation process but with tied wight matrices $W$ and $\Omega$ at each iteration step. The aggregation process in implicit GNNs is generally defined as $Z^{(l+1)} = \phi(W Z^{(l)} S + \Omega X)$ at step $l$. Given such aggregation step, implicit GNNs solve the fixed-point equation $Z^* = \phi(W Z^* S + \Omega X)$ and obtain the equilibrium $Z^*$ as node representations. To obtain the equilibrium, implicit GNNs usually require a large number of iterations of the equation until convergence, which may demand a significant amount of time. For example, MGNNI (Liu et al., 2022) ensure the convergence by using a damping factor $\gamma \in [0, 1)$ and define the aggregation as follows:

$$Z^{(l+1)} = \gamma g(W) Z^{(l)} S + f(X, \mathcal{G}), \tag{2}$$

where $g(W)$ projects the weight $W$ into a Frobenius norm ball of radius $< 1$ and $f(X, \mathcal{G})$ is a parameterized transformation. In contrast, IGNN (Gu et al., 2020) enforces $\|W\|_\infty \leq \kappa / \lambda_{pf}(A)$ with the Perron-Frobeius (PF) eigenvalue $\lambda_{pf}$ (Berman and Plemmons, 1994).

**Mini-batch sampling for GNNs** In general, mini-batch methods for GNNs need to compute the predictions for target nodes in each mini-batch. Previous mini-batch sampling methods for GNNs

generally either sample auxiliary nodes of target nodes to form a mini-batch (Hamilton et al., 2017; Zeng et al., 2021) or directly sample subgraphs from the whole graph and use a subgraph as a mini-batch (Chiang et al., 2019). For example, given a set of target nodes $\mathcal{V}_{out}$, Shadow-GNN (Zeng et al., 2021) constructs a set of auxiliary nodes $\mathcal{V}_{aux}$ by selecting nodes with top-k personalized PageRank (PPR) scores for each target node $v \in \mathcal{V}_{tgt}$. To obtain a mini-batch, it forms a subgraph $G_s$ with a node set $\mathcal{V}_s = \mathcal{V}_{tgt} \cup \mathcal{V}_{aux}$ and train GNNs on this subgraph as if it is the full graph.

# 4   SCALABLE AND EFFECTIVE IMPLICIT GNNS

As mentioned in Sec 1, previous implicit GNNs (Gu et al., 2020; Liu et al., 2022; Park et al., 2022) usually use full-batch training with an entire graph which involves recursively aggregating neighbors to calculate fixed-point solutions for nodes. It needs massive GPU memory and is not feasible for large graphs (e.g., with millions of nodes) since full-batch training requires storing a whole graph and representations of all nodes. Moreover, these works usually cost a lot of time for model training as they require a large number of iterations to solve a fixed-point equation.

Motivated by these limitations, we propose an implicit graph model SEIGNN which enables mini-batch training for implicit GNNs without harming the ability to capture long-range information and accelerates model training with a new unbiased stochastic solver.

## 4.1   MINI-BATCH TRAINING

For traditional GNNs, there are several sampling techniques proposed to improve training efficiency such as Hamilton et al. (2017); Zeng et al. (2021); Chiang et al. (2019). However, these techniques cannot be directly used to train implicit GNNs without sacrificing the accuracy, since these sampling methods enforce implicit GNNs to lose long–range dependency. To explain, these sampling-based mini-batch methods either sample neighbor nodes or a subgraph to form a training mini-batch. In this way, given a mini-batch, these methods inevitably prohibit information propagation between nodes of the current subgraph and nodes outside the subgraph. Therefore, directly using these methods can affect the advantage of implicit GNNs in capturing long-range dependencies. This is the reason why we cannot trivially use previous methods for implicit GNNs.

**Graph with coarse nodes**   To solve the above issue, inspired by graph coarsening/partitioning, we propose to use additional coarse nodes representing different partitions to facilitate long-range information propagation for implicit GNNs with mini-batch training. First, we conduct graph partitioning to obtain $k$ partitions/subgraphs $\mathcal{G}_{s_i}...\mathcal{G}_{s_k}$, which can represent the coarse information on the graph. For each partition, we create a new coarse node $v_{c_i}$ to represent the partition $\mathcal{G}_{s_i}$. We construct two types of edges as follows:

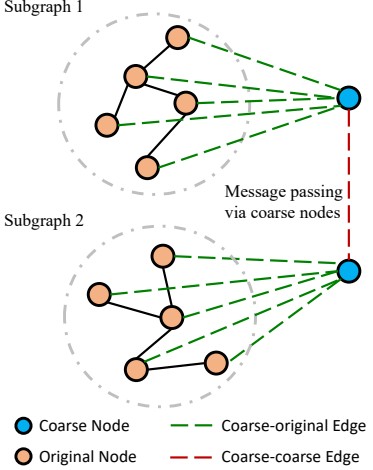

- Coarse-original edges: if a node on the original graph $v \in \mathcal{G}$ also belongs to the $i$-th partition, i.e., $v \in \mathcal{G}_{s_i}$, an edge $e = (v, v_{c_i})$ connecting $v$ and $v_{c_i}$ is constructed.

- Coarse-coarse edges: considering two different partitions $\mathcal{G}_{s_i}$ and $\mathcal{G}_{s_j}$, if there exists at least one edge $e \in \mathcal{E}$ connecting two different nodes $v_i \in \mathcal{G}_{s_i}$ and $v_j \in \mathcal{G}_{s_j}$, we construct a coarse-coarse edge connecting two corresponding coarse nodes $v_{s_i}$ and $v_{s_j}$ as $e = (v_{s_i}, v_{s_j})$.

Figure 1: Illustrations of adding coarse nodes.

With coarse-original edges, a coarse node can act as a summary of nodes in its partitions as all original nodes in this partition are its 1-hop neighbors. This can be treated as local information within each partition. Coarse-coarse edges propagate the inter-partition information by passing the message contained in a coarse node to another coarse node, which can facilitate long-range information propagation between nodes from different partitions. In addition, coarse-coarse edges also provide coarse-level graph connectivity information as global information of the whole graph.

**Mini-batch training** After constructing a new graph with additional coarse nodes, we can use existing mini-batch sampling methods, such as Shadow-GNN (Zeng et al., 2021) or GraphSAGE (Hamilton et al., 2017), to construct mini-batches for training on subgraphs. To construct a mini-batch, we first randomly sample target nodes from the node set $\mathcal{V}$, which means that we exclude coarse nodes from being target nodes. After that, to generate the prediction of each target node $v$, we choose some nodes which are more relevant or important as auxiliary nodes. As in Zeng et al. (2021), we rely on Personalized PageRank (PPR) score, a graph-structure-based metric, to determine the relevant/important nodes with respect to a target node $v$. Specifically, we select the nodes with top-k PPR scores as auxiliary nodes of the current target node $v$. Combining all target nodes and their auxiliary nodes as a node set $\mathcal{V}_s$, we obtain the subgraph $\mathcal{G}_{sub}$ with the corresponding edges connecting any two nodes in this node set.

In a mini-batch with the subgraph $\mathcal{G}_{sub}$, modifying the aggregation step in Eq (2), the fixed-point equation of our implicit GNNs can be re-written as:

$$Z^*_{sub} = \varphi(Z^*_{sub}, X, \mathcal{G}) = \gamma g(W)Z^*_{sub}S_{sub} + f(X_{sub}, \mathcal{G}_{sub}), \tag{3}$$

where $S_{sub}$ and $X_{sub}$ are the adjacency matrix and node feature matrix with only nodes in $\mathcal{G}_{sub}$.

## 4.2 Accelerate Training With New Solvers

To get the equilibrium of a fixed-point equation as node representations, previous implicit GNNs usually use an original iterative solver which simply iterates the equation (Gu et al., 2020; Liu et al., 2022; Park et al., 2022). However, it requires a large number of iterations to get the equilibrium, which costs a massive time for training. In this section, we aim to reduce the training time by reducing the number of iterations with new solvers to approximate the equilibrium of the forward pass.

**Neumann solver** First, we show that we can approximate the equilibrium with the Neumann series [2]. As proven in Liu et al. (2021), the equilibrium of Eq (3) can be obtained as follows:

$$\lim_{l \to \infty} \text{vec}[Z^{(l)}] = \text{vec}[Z^*] = (I - \gamma[S \otimes g(W)])^{-1} \text{vec}[f(X, \mathcal{G})]. \tag{4}$$

Using the fact of the Neumann series $(I - \gamma[S \otimes g(W)])^{-1} = \sum_{k=0}^{\infty} \gamma^k [S^T \otimes g(W)]^k$ and a vectorization property, we can have:

$$\text{vec}[Z^*] = \sum_{k=0}^{\infty} \gamma^k \left[ S^T \otimes g(W) \right]^k \text{vec}[f(X, \mathcal{G})] = \sum_{k=0}^{\infty} \gamma^k \text{vec}[g(W)^k f(X, \mathcal{G})S^k]. \tag{5}$$

Then, removing the vectorization from both sides, the equilibrium can be obtained as a form of infinite sum:

$$Z^* = \sum_{k=0}^{\infty} \gamma^k g(W)^k f(X, \mathcal{G})S^k. \tag{6}$$

To get the simplest approximation of the equilibrium, we can directly truncate the Neumann series at a certain step $t$ and define $V^{(t)}$ as the approximation of the equilibrium:

$$V^{(t)} = \sum_{k=0}^{t} \gamma^k g(W)^k f(X, \mathcal{G})S^k \approx Z^*. \tag{7}$$

**Stochastic solver** However, directly truncating the Neumann series to get the approximated equilibrium with Eq (7) will incur errors that are not going to vanish as the forward pass is biased. Therefore, we further propose a new **stochastic solver** to get the approximated equilibrium $\hat{Z}^*$. We first truncate the series at step $t$ as in Eq (7) and set the approximation $\hat{Z} = V^{(t)}$. At each following step $i > t$, we sample a Bernoulli random variable $b_i \sim \text{Bernoulli}(\alpha)$, which means $P(b_i = 1) = \alpha$ and $P(b_i = 0) = 1 - \alpha$. If $b_i = 1$, we update the current approximation $\hat{Z}$ with an amplifying factor $\frac{1}{\alpha^{(i-t)}}$ as follows:

$$\hat{Z}^{(i)} = \hat{Z}^{(i-1)} + \gamma^i \frac{1}{\alpha^{(i-t)}} g(W)^i f(X, \mathcal{G})S^i. \tag{8}$$

---

[2]Note that we omit the subscript "$sub$" in equations hereafter for better clarity since the equations are applicable for both a subgraph and a whole graph.

---

**Algorithm 1:** The stochastic solver's procedure for the equilibrium.

---

**Input:** The subgraph $\mathcal{G}$, the normalized adjacency matrix $S$, and the node features $X$.

**Output:** The approximated equilibrium $\hat{Z}^*$.

1   $V^{(t)} = \sum_{k=0}^{t} \gamma^k g(W)^k f(X, \mathcal{G}) S^k$;

2   $\hat{Z}^{(t)} = V^{(t)}$;

3   Define a Bernoulli distribution $p_\alpha$ where $p_\alpha[b=1] = \alpha$ and $p_\alpha[b=0] = 1 - \alpha$;

4   $i = t$ ;

5   Sample $b_i \sim p_\alpha$;

6   **while** $b_i = 1$ **do**

7     $i = i + 1$ ;

8     $\hat{Z}^{(i)} = \hat{Z}^{(i-1)} + \gamma^i \frac{1}{\alpha^{(i-t)}} g(W)^i f(X, \mathcal{G}) S^i$ ;

9     Sample $b_i \sim p_\alpha$;

10   $\hat{Z}^* = \hat{Z}^{(i)}$ ;

11   **return** $\hat{Z}^*$;

---

Otherwise if $b_i = 0$, we cease the process and obtain the final approximation as $\hat{Z}^* = \hat{Z}^{(i)}$. The procedure of our stochastic solver is illustrated in Algorithm 1.

Using the stochastic solver, we can obtain the unbiased approximation of the equilibrium as shown in the following proposition.

**Proposition 1.** *The proposed stochastic solver is an unbiased estimator of the equilibrium $Z^*$ of the forward pass, i.e., the expectation of the approximated equilibrium $\hat{Z}^*$ is the same as that of the true equilibrium $Z^*$: $\mathbb{E}[\hat{Z}^*] = Z^* = \sum_{k=0}^{\infty} \gamma^k g(W)^k f(X, \mathcal{G}) S^k$, under the condition that $\sum_{k=t+1}^{\infty} \gamma^k g(W)^k f(X, \mathcal{G}) S^k \frac{1}{\alpha^{k-t}}$ exists* [3].

We provide the proof of this proposition in Appendix A.1. Our stochastic solver is an **unbiased stochastic solver** which can have the same error with fewer iterations in expectation compared with our Neumann solver and the original iterative solver (Gu et al., 2020) used in existing implicit GNNs.

### 4.3   TRAINING OF SEIGNN

For model training, with a subgraph in a mini-batch, we use our unbiased stochastic solver with Eq (8) to obtain the approximated equilibrium $\hat{Z}^*$ for the forward pass. For backward pass, as shown in Liu et al. (2022) and Bai et al. (2019), implicit differentiation is used to compute the gradients by directly differentiating through the equilibrium as:

$$\frac{\partial \ell}{\partial (\cdot)} = \frac{\partial \ell}{\partial \hat{Z}^*} \left( I - J_\varphi(\hat{Z}^*) \right)^{-1} \frac{\partial \varphi(\hat{Z}^*, X, \mathcal{G})}{\partial (\cdot)}, \tag{9}$$

where $\hat{Z}^* = \varphi(\hat{Z}^*, X, \mathcal{G})$ is the fixed-point equation and $J = \frac{\partial \varphi(\hat{Z}^*, X, \mathcal{G})}{\partial \hat{Z}^*}$. To avoid expensive computation of calculating $\left( I - J_\varphi(\hat{Z}^*) \right)^{-1}$, we adopt a recently proposed phantom gradient estimation (Geng et al., 2021) which has the advantages on efficient computation and stable training dynamics. Therefore, with our unbiased stochastic solver and phantom gradient estimation, we can enjoy efficient computation for both forward and backward passes.

## 5   EXPERIMENTS

In this section, we demonstrate the effectiveness and efficiency of SEIGNN compared with both implicit GNNs and representative traditional GNNs on large graph datasets for the node classification task. Specifically, we conduct experiments on 6 commonly used datasets for node classification (i.e., Flickr, Yelp, Reddit, PPI, ogbn-arxiv, and ogbn-products). We provide the descriptions and details of datasets in Appendix B.1.

---

[3] We discuss the case when this condition does not hold in Appendix A.1.

Table 1: Comparison on test accuracy / micro F1 score on large graph datasets.

| Model | Flickr | Reddit | Yelp | PPI |
|---|---|---|---|---|
| GCN (Kipf and Welling, 2016) | $49.2 \pm 0.3$ | $93.3 \pm 0.1$ | $37.8 \pm 0.1$ | $51.2 \pm 0.3$ |
| GraphSAGE (Hamilton et al., 2017) | $50.1 \pm 1.3$ | $95.3 \pm 0.1$ | $63.4 \pm 0.6$ | $63.7 \pm 0.6$ |
| FastGCN (Chen et al., 2018) | $50.4 \pm 0.1$ | $92.4 \pm 0.1$ | $26.5 \pm 5.3$ | $51.3 \pm 3.2$ |
| ASGCN (Huang et al., 2018) | $50.4 \pm 0.2$ | $95.8 \pm 0.1$ | $63.4 \pm 0.6$ | $68.7 \pm 1.2$ |
| ClusterGCN (Chiang et al., 2019) | $48.1 \pm 0.5$ | $95.4 \pm 0.1$ | $60.9 \pm 0.5$ | $89.5 \pm 0.4$ |
| GraphSAINT (Zeng et al., 2020) | $51.5 \pm 0.1$ | $96.7 \pm 0.1$ | $64.5 \pm 0.3$ | $98.0 \pm 0.2$ |
| IGNN (Gu et al., 2020) | $53.0 \pm 0.2$ | $97.0 \pm 0.2$ | $65.8 \pm 0.2$ | $97.8 \pm 0.1$ |
| MGNNI (Liu et al., 2022) | $53.5 \pm 0.2$ | $96.2 \pm 0.2$ | $56.6 \pm 0.1$ | $98.6 \pm 0.1$ |
| USP (Li et al., 2023) | $54.3 \pm 0.1$ | $96.8 \pm 0.2$ | $66.1 \pm 0.2$ | $98.5 \pm 0.2$ |
| SEIGNN | $\mathbf{55.8 \pm 0.1}$ | $\mathbf{97.8 \pm 0.1}$ | $\mathbf{66.9 \pm 0.2}$ | $\mathbf{98.8 \pm 0.2}$ |

Table 2: Test Accuracy on OGBN datasets.

| Model | ogbn-arxiv | ogbn-products |
|---|---|---|
| IGNN | $70.4 \pm 0.8$ | $69.7 \pm 0.8$ |
| MGNNI | $71.2 \pm 0.4$ | OOM |
| USP | $72.7 \pm 0.2$ | $73.6 \pm 0.3$ |
| SEIGNN | $\mathbf{77.9 \pm 0.2}$ | $\mathbf{76.4 \pm 0.2}$ |

Table 3: Training time per epoch (second).

| Model | Flickr | Reddit | ogbn-arxiv |
|---|---|---|---|
| IGNN | 7.68 | 141.12 | 8.51 |
| MGNNI | 9.29 | 60.95 | 4.04 |
| USP | 3.13 | 52.78 | 7.05 |
| SEIGNN | 4.36 | 6.21 | 3.79 |

## 5.1 NODE CLASSIFICATION

**Comparison on popular large graphs** We first use the various graph datasets with relatively large sizes (i.e., Flickr, Reddit, Yelp, PPI). Their scenarios vary from predicting communities of online posts (Reddit) to classifying protein functions (PPI).

The results are shown in Table 1. We can see that SEIGNN generally outperforms all other representative baselines including both implicit GNNs and traditional GNNs on these four datasets. Compared with USP (Li et al., 2023) which uses mini-batch training, SEIGNN achieves better performance by up to 1.5% absolute improvement. This can be attributed to the better ability of SEIGNN to capture global and long-range information by adding coarse nodes during training. IGNN and MGNNI, two implicit GNNs with full-batch training, generally perform worse than USP and SEIGNN, indicating that mini-batch training is more effective on large graphs in terms of performance. In addition, as MGNNI shares a similar aggregation step as SEIGNN, the worse performance of MGNNI compared with SEIGNN verifies the effectiveness of our designs of mini-batch training with the stochastic solver and coarse nodes.

**Comparison on OGBN datasets** Apart from four large graph datasets, to better examine scalability and effectiveness, we also conduct experiments with two popular OGBN datasets (ogbn-arxiv and ogbn-products). Specifically, ogbn-products is the largest dataset used in this paper, which contains around 2.5 million nodes and 61 million edges.

Table 2 shows the comparison of accuracies between SEIGNN and other implicit GNNs on OGBN datasets. SEIGNN can still outperform other implicit GNNs by a large margin. Specifically, SEIGNN achieves 5.1% and 2.7% absolute accuracy improvements on ogbn-arxiv and ogbn-products respectively. In addition, we observe that MGNNI would face the out-of-memory (OOM) issue on ogbn-products as MGNNI has to load all nodes into GPU memory for full-batch training. This verifies again the limitation of using full-batch training in previous implicit GNNs.

**Efficiency Comparison** In addition to evaluating prediction accuracy, we also provide experimental results regarding the training efficiency of different implicit GNN models. Table 5 demonstrates the comparison of training time per epoch among different models. We can see that our model SEIGNN generally has less training time per epoch compared with existing implicit GNNs. Especially, on

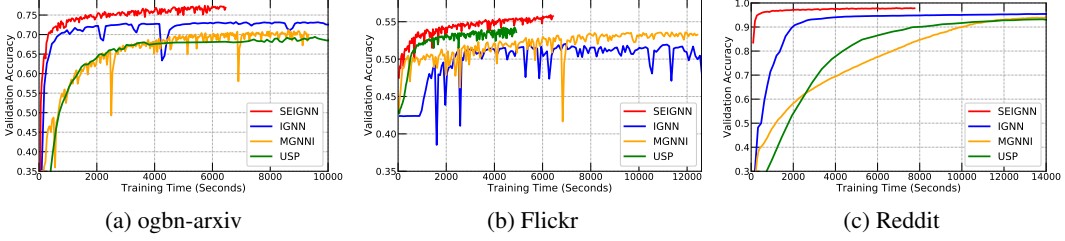

Figure 2: Efficiency Comparison: Accuracy vs Training Time.

Table 4: Accuracy comparison: the effectiveness of using coarse nodes and the unbiased stochastic solver.

| Model | Reddit | Yelp | ogbn-arxiv |
|---|---|---|---|
| SEIGNN (full version) | 97.85 | 66.85 | 77.94 |
| w/o coarse nodes | 96.34 | 63.70 | 71.40 |
| remove stochastic solver | 97.51 | 66.65 | 77.24 |

Table 5: Accuracy of different mini-batch methods.

| Method | ogbn-arxiv |
|---|---|
| ClusterGCN | 71.31 |
| GraphSAGE | 68.84 |
| SEIGNN | 77.94 |

Reddit, SEIGNN only needs 6.21s for an epoch, which is around 8x less compared with USP. Note that, although USP spends the least time training an epoch on Flickr which is relatively small, it requires more time compared with SEIGNN on other large datasets. This may indicate that USP is not efficient enough for relatively large and dense datasets (e.g., Reddit).

Moreover, for a more holistic view of efficiency comparison, in Figure 2, we show how accuracy would change as training progresses for different models. First of all, our model SEIGNN achieves better eventual accuracies with less total training time compared with other implicit GNNs. Additionally, the accuracy of SEIGNN also increases much faster. These observations confirm the effectiveness and efficiency of our model on large graphs.

Besides the efficiency comparison, we also provide the comparison of memory usage in Table 9 of Appendix B.3, showing that SEIGNN has significantly less GPU memory usage compared with other implicit GNNs. Lower memory usage leads to higher scalability of our model.

## 5.2 ABLATION STUDY AND FURTHER EXPERIMENTAL INVESTIGATION

Besides the overall performance and efficiency comparison in the above section, we also conduct detailed ablation studies and further investigations about the effectiveness of two key components in SEIGNN (i.e., mini-batch training with coarse nodes and the unbiased stochastic solver).

**Effectiveness of using coarse nodes and the proposed solver** Table 4 demonstrates the results of removing coarse nodes in our designed mini-batch method and replacing the unbiased stochastic solver with the naive Neumann solver. It shows that, without adding coarse nodes in mini-batch training, the accuracies drop significantly, especially on Yelp and ogbn-arxiv. This confirms that adding coarse nodes is helpful for better global information propagation. By replacing our unbiased stochastic solver with the naive Neumann solver, the performance also slightly decreases, which indicates the effectiveness of the unbiased stochastic solver.

**Ineffectiveness of directly applying existing mini-batch methods** In Section 1 and 4.1, we explain the reason that directly applying existing mini-batch methods for implicit GNNs may affect the performance. Table 5 empirically verifies the ineffectiveness of using existing mini-batch methods by illustrating that trivially using those methods (i.e., ClusterGCN (Chiang et al., 2019) and GraphSAGE with neighbor sampling (Hamilton et al., 2017)) for implicit GNNs provides much worse performances compared to our mini-batch training method with coarse nodes.

Table 6: Compatibility: using coarse nodes can improve other existing mini-batch methods.

| Method | Variant | Accuracy |
|---|---|---|
| ClusterGCN | w/o coarse nodes | 71.31 |
| | w/ coarse nodes | 76.21 |
| GraphSAGE | w/o coarse nodes | 68.84 |
| | w/ coarse nodes | 73.89 |

Table 7: Accuracy and Total Time (second) of different solvers with different numbers of maximum iterations on ogbn-arxiv.

| Max iter. | our solver 3 | original solver 5 | 10 | 50 |
|---|---|---|---|---|
| Accuracy | 77.94 | 77.19 | 77.53 | 77.79 |
| Total Time | 7891 | 15399 | 21748 | 45775 |

**The improvements are higher for low-degree nodes.** As the significant improvement by using coarse nodes is shown in Table 4, we try to provide more insights by further investigating how coarse nodes may specifically improve the performance. Considering nodes with different levels of degrees, we evenly split nodes into 5 groups according to their degrees (1st group contains nodes with the highest level of degrees while the 5th group contains the lowest-degree nodes). Figure 3a shows the average accuracies of different degree groups by comparing the variants using/not using coarse nodes, and Figure 3b demonstrates the relative improvement of degree groups through adding coarse nodes. We can see that 1) nodes with lower degrees tend to have low accuracies for both two variants, and 2) accuracy improvements on nodes with lower degrees are more obvious compared with nodes with higher degrees. These observations suggest that our proposed mini-batch training with coarse nodes is more helpful on nodes with lower degrees. The reason might be that, by enhancing global/long-range information propagation via coarse nodes, low-degree nodes can receive sufficient information compared with the variant not adding coarse nodes.

Additionally, we also investigate the compatibility of using coarse nodes for applying other sampling-based mini-batch methods to SEIGNN. Table 6 shows that, on ogbn-arxiv, adding coarse nodes can also improve the performance of two existing mini-batch methods (i.e., ClusterGCN and GraphSAGE with neighbor sampling).

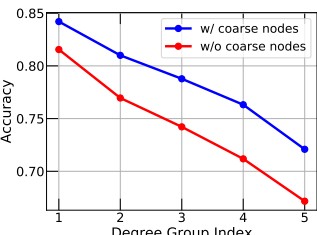

(a) Accuracy comparison for different degree groups.

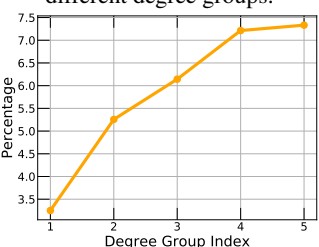

(b) Relative Improvement.

Figure 3: The effect of using coarse nodes for nodes with different levels of degrees on ogbn-arxiv.

**Efficiency comparison for different solvers** Moreover, to verify the effectiveness of our stochastic solver, in Table 7, we provide an experimental comparison of accuracy and total time between our solver and the original solver used in (Liu et al., 2022; Gu et al., 2020). We use different maximum iterations for the original solvers and set maximum iterations as 3 for our solver with the continue probability $\alpha = 0.5$. The results show that our solver can generally achieve better accuracy compared with the original solver while spending much less time. We also observe that the original solver needs more iterations (i.e., 50) to achieve a comparable accuracy as our solver, which leads to excessive time consumption.

## 6 CONCLUSION

In this paper, we propose a scalable and effective implicit GNN model (SEIGNN) that can be efficiently trained on large graphs. Specifically, SEIGNN contains a mini-batch training method with added coarse nodes and an unbiased stochastic solver to scale up the model to large graphs without losing the ability to capture long-range information. The experiments on several large-graph datasets demonstrate that SEIGNN can achieve superior performance using less training time compared with existing implicit GNNs. Furthermore, the results of ablation studies verify the effectiveness of our mini-batch training method and the unbiased stochastic solver. We also try to provide a deeper analysis of why using coarse nodes in our mini-batch training can improve performance.

## ACKNOWLEDGMENTS

This research is supported by the Ministry of Education, Singapore, under its MOE AcRF TIER 3 Grant (MOE-MOET32022-0001). Any opinions, findings and conclusions or recommendations expressed in this material are those of the author(s) and do not reflect the views of the Ministry of Education, Singapore.

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

# Appendices

## A   PROOFS

### A.1   PROOF OF PROPOSITION 1

*Proof.* We first define $S_i \in \{0, 1\}$ as a random variable, indicating whether we sample the $i$-th term or not. By the property of probability, we can decompose the marginal $\mathbb{P}[S_t = 1]$ by

$$\mathbb{P}(S_i = 1) = \mathbb{P}[S_i = 1|S_{i-1} = 1] \cdot \mathbb{P}[S_{i-1} = 1] + \mathbb{P}[S_i = 1|S_{i-1} = 0] \cdot \mathbb{P}[S_{i-1} = 0].$$

Here, $\mathbb{P}[S_i = 1|S_{i-1} = 0] = 0$ in our algorithm because we terminate the iteration if $S_{i-1} = 0$. Thus,

$$\mathbb{P}[S_i = 1] = \mathbb{P}[S_i = 1|S_{i-1} = 1] \cdot \mathbb{P}[S_{i-1} = 1].$$

Moreover, as the random variable $S_i$ is sampled from a Bernoulli distribution $Bernoulli(\alpha)$, we have $\mathbb{P}[S_i = 1|S_{i-1} = 1] = \alpha$ in our algorithm. Plugging this into the above,

$$\mathbb{P}[S_i = 1] = \alpha \mathbb{P}[S_{i-1} = 1].$$

Recursively applying this equation,

$$\mathbb{P}[S_i = 1] = \alpha^{i-t} \mathbb{P}[S_t = 1].$$

Since $\mathbb{P}[S_i = 1] = 1$ for all $i \leq t$ in our algorithm, we have

$$\mathbb{P}[S_i = 1] = \alpha^{i-t}.$$

Considering the possibility of reaching step $i$, the forward pass of our algorithm can be written as:

$$\hat{Z}^{(i)} = \hat{Z}^{(t)} + \sum_{k=t+1}^{i} \gamma^k \frac{\mathbb{1}\{S_k = 1\}}{\alpha^{(k-t)}} g(W)^k f(X, \mathcal{G}) S^k,$$

where $\mathbb{1}\{S_k = 1\}$ take the value 1 when $S_k = 1$ (i.e., the algorithm reaches step $k$), otherwise it takes the value 0.

As we continue the algorithm until the termination (i.e., $S_i = 0$), we can write the approximated equilibrium as follows:

$$\hat{Z}^* = \hat{Z}^{(t)} + \sum_{k=t+1}^{\infty} \gamma^k \frac{\mathbb{1}\{S_k = 1\}}{\alpha^{(k-t)}} g(W)^k X S^k.$$

By taking the expectation of the second term, with the definition $A_i = \gamma^i g(W)^i f(X, \mathcal{G}) S^i$ and the condition that $\sum_{k=t+1}^{\infty} A_k \frac{1}{\alpha^{k-t}}$ exists, we have the following:

$$
\begin{aligned}
\mathbb{E}\left[\sum_{k=t+1}^{\infty} \gamma^k \frac{\mathbb{1}\{S_k = 1\}}{\alpha^{(k-t)}} g(W)^k X S^k\right] &= \mathbb{E}\left[\sum_{k=t+1}^{\infty} A_k \frac{\mathbb{1}\{S_k = 1\}}{\alpha^{k-t}}\right] \\
&= \sum_{k=t+1}^{\infty} A_k \mathbb{E}\left[\frac{\mathbb{1}\{S_k = 1\}}{\alpha^{k-t}}\right] \\
&= \sum_{k=t+1}^{\infty} A_k \frac{\mathbb{P}[S_k = 1]}{\alpha^{k-t}} \\
&= \sum_{k=t+1}^{\infty} A_k = \sum_{k=t+1}^{\infty} \gamma^k g(W)^k f(X, \mathcal{G}) S^k,
\end{aligned}
$$

Combining the above expectation of the second term with the deterministic term $\hat{Z}^{(t)} = \sum_{i=0}^{t} \gamma g(W)^k f(X, \mathcal{G}) S^k$, we get obtain the expectation $\mathbb{E}[\hat{Z}^*]$ as follows:

$$\mathbb{E}[\hat{Z}^*] = \hat{Z}^{(t)} + \mathbb{E}\left[\sum_{k=t+1}^{\infty} \gamma^k \frac{\mathbb{1}\{S_k = 1\}}{\alpha^{(k-t)}} g(W)^k X S^k\right]$$

$$= \sum_{k=0}^{t} \gamma^k g(W)^k f(X, \mathcal{G}) S^k + \sum_{k=t+1}^{\infty} \gamma^k g(W)^k f(X, \mathcal{G}) S^k$$

$$= \sum_{k=0}^{\infty} \gamma^k g(W)^k f(X, \mathcal{G}) S^k = Z^*.$$

This indicates that our proposed stochastic solver is an unbiased estimator of the equilibrium $Z^*$. □

In the above proof, we have an assumption that $\sum_{k=t+1}^{\infty} A_k \frac{1}{\alpha^{k-t}}$ exists. In the case where $\sum_{k=t+1}^{\infty} A_k \frac{1}{\alpha^{k-t}}$ does not exist, we define $f_n$ to be the output of a modified version of the Algorithm 1 where we replace the while-loop with the for-loop up to $n$ step: i.e., we forceful terminate the while-loop if it takes more than $n$ steps. Then, by using the same proof steps except that we replace the infinite sum with the finite sum upto $n$ terms, we conclude that for any $n$, $\mathbb{E}[f_n] = \sum_{k=0}^{n} A_k$. This implies that our proposed stochastic solver is still an unbiased estimator of the equilibrium $Z^*$ of the forward pass up to the error $\sum_{k=n}^{\infty} A_k$. For any desired error value $\epsilon > 0$ (including machine precision), there exists a sufficiently large $n$ such that $\sum_{k=n}^{\infty} A_k \leq \epsilon$. Thus, the statement in Proposition 1 still holds true up to the machine precision without the condition that $\sum_{k=t+1}^{\infty} A_k \frac{1}{\alpha^{k-t}}$ exists. The output of our algorithm is equivalent to $f_n$ with $n = \infty$. Thus, it is ensured to be unbiased.

## B  MORE ON EXPERIMENTS

### B.1  DATASET STATISTICS AND DESCRIPTIONS

The dataset statistics are provided in Table 8. ogbn-products is the largest dataset used in our paper, which contains around 25 million nodes and 61 million edges. Reddit is the densest dataset here with the average node degrees as 50. We follow Li et al. (2023) to use six datasets in our experiments. We provide a detailed description of each dataset as follows:

- **Flickr** is a single-label multi-class classification dataset. The task is to categorize types of images based on the descriptions and common properties of online images. We are using the Flickr dataset as provided in Zeng et al. (2020). Flickr data are collected in the SNAP website [4] from different sources. Flickr contains an undirected graph and a node in the graph represents an image on Flickr. An edge is connected between two nodes if two corresponding images share some common properties (e.g., the same gallery, comments from the same user, etc.). The node features are the 500-dimensional bag-of-word representations of the images. For labels, each image belongs to one of the 7 classes.

- **Reddit** is a single-label multi-class classification dataset. The task is to predict different communities of online posts. We use the Reddit dataset from Hamilton et al. (2017) as in Li et al. (2023). The nodes are online posts and an edge is connected between two posts if the same user comments on both. Word features in posts are 300-dimensional word vectors. Node features are concatenated using 1) the average embedding of the post title, 2) the average embedding of all the post's comments, 3) the post's score, 4) the number of comments on the post.

- **Yelp** is a multi-label multi-class classification dataset. The task is to categorize types of businesses based on users and friendships. We use the Yelp dataset provided in Zeng et al. (2020). Yelp contains a single graph. The nodes are users who provide reviews. If two users are friends, an edge between them is connected. The features of each node are added and normalized using several 300-dimensional vectors representing a review word provided by the user.

---

[4]https://snap.stanford.edu/data/web-flickr.html

Table 8: Dataset statistics.

| Dataset | Nodes | Edges | Avg. Degree | Classes | Features | Train/Val/Test |
|---|---|---|---|---|---|---|
| Flickr | 89,250 | 899,756 | 10 | 7 | 500 | 0.50/0.25/0.25 |
| Reddit | 232,965 | 11,606,919 | 50 | 41 | 602 | 0.66/0.10/0.24 |
| Yelp | 716,847 | 6,977,410 | 10 | 100 | 300 | 0.75/0.10/0.15 |
| PPI | 14,755 | 225,270 | 14 | 121 | 50 | 0.79/0.11/0.10 |
| ogbn-arxiv | 169,343 | 1,166,243 | 7 | 40 | 128 | 0.54/0.18/0.29 |
| ogbn-products | 2,449,029 | 61,859,140 | 25 | 100 | 47 | 0.10/0.02/0.88 |

- **PPI** is a single-label multi-class classification dataset that contains multiple graphs. The task is to classify protein functions based on the interactions of human tissues. PPI dataset has 24 graphs in total and each graph represents a different human tissue. In a graph, nodes represent proteins and edges indicate interactions between proteins. Each node can have up to 121 labels, which are originally collected from Molecular Signatures dataset (Subramanian et al., 2005) by Hamilton et al. (2017). The dataset splits used in our paper are the same as in Hamilton et al. (2017), i.e., 20 graphs for training, 2 graphs for validation, and 2 graphs for testing.

- **ogbn-arxiv** is a single-label classification dataset that contains a directed graph. The task is to predict the 40 subject areas of arXiv CS papers, such as cs.AI, cs.LG, and cs.OS, which are manually labeled by the authors of the paper and the moderators. each node is an arXiv paper and each directed edge indicates that one paper cites another one. The node feature of each node is a 128-dimensional vector obtained by averaging the embeddings of words in the title and abstract. We download ogbn-arxiv dataset from the OGB website [5]. The detailed descriptions can be found in Hu et al. (2020).

- **ogbn-products** is a single-label multi-class classification dataset which contains an undirected and unweighted graph. Nodes represent products sold on Amazon, and edges between two products indicate that these two products are purchased together. Node features are generated by extracting bag-of-words features from the description of the product followed by a Principal Component Analysis to reduce the dimension to 100. We download ogbn-products dataset from the OGB website [5]. The detailed descriptions can be found in Hu et al. (2020).

## B.2 EXPERIMENTAL SETTING

For experimental setup, we mainly follow Li et al. (2023). As we use the same experimental setting on some datasets, we reuse the results of some baselines from Li et al. (2023) and Zeng et al. (2020). We compare SEIGNN with 3 implicit GNNs (i.e., USP (Li et al., 2023), MGNNI (Liu et al., 2022), and IGNN (Gu et al., 2020)) and 6 explicit/traditional GNNs (GCN (Kipf and Welling, 2016), GraphSAGE (Hamilton et al., 2017), FastGCN (Chen et al., 2018), ASGCN (Huang et al., 2018), ClusterGCN (Chiang et al., 2019), and GraphSAINT (Zeng et al., 2020)). The experiments are run with 5 different trials. The averaged accuracy and standard deviation are reported. We mainly run the experiments on an RTX-A5000 GPU with 24GB GPU memory.

**Model Architecture and Hyperparameters** For SEIGNN, we use the same structure with a few implicit graph layers and the same number of linear layers as in MGNNI (Liu et al., 2022) and USP (Li et al., 2023). We select the number of implicit graph layers from {2, 3, 4}. We also conduct a hyperparameter search on learning rate {0.01, 0.005, 0.001} and dropout rate {0.0, 0.2, 0.5}. The number of deterministic steps $t$ in our stochastic solver is chosen from {3, 5} and the continuation probability $\alpha$ is set to 0.5. The hyperparameter $\gamma$ used in an implicit graph layer is set to 0.8. The Adam optimizer (Kingma and Ba, 2015) is used for optimization. The number of partitions for adding coarse nodes in our mini-batch training method is selected from {50, 100, 200}. The number of target nodes in a mini-batch is configured as follows: 8192 for Flickr and PPI, 10240 for ogbn-arxiv, Yelp, and Reddit, and 16384 for ogbn-products.

---

[5] https://ogb.stanford.edu/docs/nodeprop

Table 9: GPU Memory Usage (GB) of different models. OOM indicates Out-of-Memory.

| | 2-layer | | | 3-layer | | |
| --- | --- | --- | --- | --- | --- | --- |
| | SEIGNN | MGNNI | IGNN | SEIGNN | MGNNI | IGNN |
| Reddit | 7.23 | 23.77 | 18.67 | 8.41 | OOM | 22.21 |
| Yelp | 5.82 | 15.96 | 13.65 | 7.27 | 20.72 | 17.34 |
| ogbn-products | 9.21 | OOM | OOM | 9.78 | OOM | OOM |

### B.3 ADDITIONAL EXPERIMENTAL RESULTS

Besides the overall comparison of accuracy and efficiency, we also investigate the memory usage of different implicit GNNs. Table 9 shows that SEIGNN requires significantly less GPU memory compared with IGNN and MGNNI. In particular, SEIGNN only uses 37% of the memory as IGNN with 3 implicit layers on Reddit. Moreover, we can see that SEIGNN cost less than 10GB GPU memory on ogbn-products while MGNNI and IGNN face the out-of-memory issue using a GPU with 24GB memory.

