# OpenReview forum: "Scalable and Effective Implicit Graph Neural Networks on Large Graphs"
_ICLR.cc/2024/Conference — ICLR 2024 poster_

### Official Review · Reviewer_m8F2 · 2023-10-16

**Soundness:** 3 good
**Presentation:** 4 excellent
**Contribution:** 3 good
**Rating:** 8
**Confidence:** 4

**Summary:**

This work aims to address the scalability and efficiency challenges associated with implicit Graph Neural Networks (GNNs), which are designed to capture long-range dependencies in graph-structured data. It first points out that the implicit GNNs are suffering from the computational burden brought by (1) full-batch training and (2) a large number of iterations to solve fixed-point equations, limiting their applicability to large graphs. This work proposes a scalable and effective SEIGNN model that employs mini-batch training with coarse-level nodes to encourage information propagation between subgraphs. Extensive experiments are conducted to justify the power of SEIGNN in terms of both efficiency and efficacy.

**Strengths:**

1. This work is clearly motivated, addressing the scalability issue of IGNN is an important problem.

2. Good performance and comprehensive ablation study.

3. Writing is clear and easy to follow.

**Weaknesses:**

Please refer to ``Questions``.

**Questions:**

1. I agree that subgraph-wise sampling (ClusterGCN) and node-wise sampling (GraphSage) indeed lead to a loss of long-range dependencies. However, how about layer-wise sampling, such as the approaches like LADIES[1] and fastGCN[2]?

2. I want to confirm my understanding. In subgraph sampling, the coarse nodes are excluded from being the target nodes but will be included in the sampled subgraph through the top-k Personalized PageRank (PPR) selection step, is that correct?

3. Given that the purpose of introducing the coarse nodes is to maintain long-term dependencies in mini-batch training, can SEIGNN ensure that coarse nodes are included in each mini-batch? Or is there a way to encourage the sampled subgraph to include more coarse nodes?


[1] https://arxiv.org/abs/1911.07323
[2] https://arxiv.org/abs/1801.10247

---

> ### Author Response · Authors · 2023-11-21
> **Response to Reviewer m8F2**
>
> We appreciate reviewer m8F2 for the valuable feedback and suggestions. We would like to clarify the questions as below.
>
> > **Q1:** I agree that subgraph-wise sampling (ClusterGCN) and node-wise sampling (GraphSage) indeed lead to a loss of long-range dependencies. However, how about layer-wise sampling, such as the approaches like LADIES[1] and fastGCN[2]?
>
> For layer-wise sampling, it would still lose the ability to capture long-range dependencies as node-wise sampling. The major difference between layer-wise and node-wise sampling is that layer-wise sampling fixes the number of overall sampled nodes **in a layer** to a certain number while node-wise sampling fixes the number of sampled neighbors for **each node**. Thus, layer-wise sampling may have even more restricted sampled minibatches and face a similar problem of losing long-range dependencies as node-wise sampling.
>
> > **Q2:** I want to confirm my understanding. In subgraph sampling, the coarse nodes are excluded from being the target nodes but will be included in the sampled subgraph through the top-k Personalized PageRank (PPR) selection step, is that correct?
>
> Yes. Coarse nodes will be included in the sampled subgraph through top-k PPR selection.
>
> > **Q3:** Given that the purpose of introducing the coarse nodes is to maintain long-term dependencies in mini-batch training, can SEIGNN ensure that coarse nodes are included in each mini-batch? Or is there a way to encourage the sampled subgraph to include more coarse nodes?
>
> Currently, SEIGNN does not directly force coarse nodes to be included in each mini batch. However, coarse nodes can be selected as they may have higher Personalized PageRank (PPR) scores. Which auxiliary nodes are selected is determined by PPR scores, indicating the more relevant nodes are more likely to be selected.
>
> Specifically for coarse nodes, it's also possible to encourage the sampled subgraph to have more coarse nodes. For example, we directly add all coarse nodes that are 1-hop neighbors of the target nodes into the sampled subgraph. Or more extremely, we can even force all coarse nodes included in the mini-batch when global information is very important for a specific dataset. Although it is interesting to explore further, we leave it as future work due to time limitations.
>
> ---
> Thank you again for your valuable feedback. Hope our response answers your questions.

---

### Official Review · Reviewer_K2Aq · 2023-10-26

**Soundness:** 3 good
**Presentation:** 3 good
**Contribution:** 2 fair
**Rating:** 3
**Confidence:** 5

**Summary:**

The paper proposes a scalable and effective implicit graph neural network (GNN) model, called SEIGNN, that can handle large graph data. Implicit GNNs are models that can capture long-range dependencies in graphs by solving a fixed-point equation. However, they are computationally expensive and memory-intensive, due to their use of full-batch training and a large number of iterations. SEIGNN addresses these limitations by using a mini-batch training method with coarse nodes and a stochastic solver. The coarse nodes are added to the mini-batch to incorporate global information from the graph, and the stochastic solver is used to obtain unbiased approximate solutions with fewer iterations. SEIGNN can be integrated with any existing implicit GNN model to improve its performance. The paper evaluates SEIGNN on six datasets and shows that it achieves better accuracy with less training time than existing implicit GNNs.

**Strengths:**

* The paper tackles a novel and important problem of scaling up implicit GNNs to large graphs, which has applications in real-world massive graphs.
* The authors introduce a mini-batch sampling strategy that can preserve the global and long-range information of the graph by using coarse nodes and can achieve reduced training time by using a stochastic solver.
* Extensive experiments and analysis are conducted to demonstrate the advantages of the proposed framework in mini-batch training.

**Weaknesses:**

* The paper fails to position itself with recent related works. More SOTA papers in mini-batch training strategy should be included for discussion, such as [1-3]. The idea of introducing coarse nodes during the process of mini-batching is relatively straightforward and appears in existing literature such as [3]. Therefore, it hurts the novelty of this work regarding the first contribution without a thorough comparison with the current mini-batching GNN methods that also include coarse nodes. Meanwhile, there is a very relevant and similar work that should be added for comparison [4].

* For the second contribution of accelerated training with the Neumann solver, it is unclear how the authors constrain $n$ to be sufficiently large during experiments. In other words, does the performance of SEIGNN be affected even if the proposed stochastic solver is not an unbiased estimator? Could the authors elaborate more on this?

* The setting of the experiments could be improved to better reflect the contributions. Do the results for all implicit GNNs come from full-batch training on all benchmarks? What would performance be if we incorporated different mini-batching sampling strategies for the baseline implicit GNNs? Meanwhile, more recent baselines of scalable GNN should be added for comparison such as [5-6].

* This manuscript aims to scale implicit GNNs to large graphs, but the benchmarks have nearly become standard for even some GNNs that are not specifically designed for scalability. Therefore, large graphs such as ogbn-papers100M should be evaluated for the scalability aim. Note that even the largest dataset (ogbn-products) used in this manuscript is marked as medium in OGB benchmarks.

[1] VQ-GNN: A Universal Framework to Scale-up Graph Neural Networks using Vector Quantization, NeurIPS 2021

[2] Influence-Based Mini-Batching for Graph Neural Networks, LoG 2022

[3] SMGRL: Scalable Multi-resolution Graph Representation Learning, arXiv 2022

[4] Efficient and Scalable Implicit Graph Neural Networks with Virtual Equilibrium, Big Data 2022

[5] Decoupling the Depth and Scope of Graph Neural Networks, NeurIPS 2021

[6] Sketch-GNN: Scalable Graph Neural Networks with Sublinear Training Complexity, NeurIPS 2022

**Questions:**

Please kindly refer to the Weaknesses.

---

> ### Author Response · Authors · 2023-11-21
> **Response to Reviewer K2Aq [1/2]**
>
> [Part1 / 2] We appreciate reviewer K2Aq for the valuable feedback and suggestions. However, we respectfully disagree with some comments and we believe there might be some misunderstandings. We would like to clarify and answer the questions as below.
>
> ---
> > **W1:** The paper fails to position itself with recent related works.
>
> First, we would like to emphasize again that our submission mainly focuses on scalable **implicit** GNNs instead of scalable traditional/explicit GNNs. Therefore, for scalable traditional/explicit GNNs, we only cover/discuss some well-known sampling-based works such as GraphSAGE, ClusterGCN, GraphSAINT as in USP [1].
>
> For VQ-GNN [2], it uses quantization to scale up GNNs, which is a different direction for improving scalability compared with sampling-based works.
>
> For SMGRL [3], in our humble opinion, it actually does not add coarse nodes for training. It directly trains on different coarse graphs rather than considering both original nodes and coarse graphs for training simultaneously. Additionally SMGRL is for explicit GNNs, which is not our main focus.
>
> For VEQ [4] which is published in Big Data conference 2022, we agree that it is relevant and we apologize that we did not notice as it is not publicly available on Arxiv (conference proceedings are also not available without purchase).
> Comparing VEQ and our work SEIGNN, VEQ utilizes previous equilibrium states as an initialization for speeding up training while our work proposed a new solver to obtain unbiased approximation of equilibrium. For mini-batch training, VEQ has specific designs for update while our work SEIGNN is compatible with different sampling methods as shown in Table 6. Moreover, in terms of accuracy, we observe that SEIGNN is better than VEQ on some datasets such as Flickr, Reddit, Yelp, and ogbn-arxiv. We will include above discussions/comparisons into a later version of our manuscript.
>
> [1] Unbiased Stochastic Proximal Solver for Graph Neural Networks with Equilibrium States, ICLR 2023.
>
> [2] VQ-GNN: A Universal Framework to Scale-up Graph Neural Networks using Vector Quantization, NeurIPS 2021.
>
> [3] SMGRL: Scalable Multi-resolution Graph Representation Learning, arXiv 2022.
>
> [4] Efficient and Scalable Implicit Graph Neural Networks with Virtual Equilibrium, Big Data 2022.
>
> ---
> > **W2:** For the second contribution of accelerated training with the Neumann solver, it is unclear how the authors constrain
> to be sufficiently large during experiments. In other words, does the performance of SEIGNN be affected even if the proposed stochastic solver is not an unbiased estimator? Could the authors elaborate more on this?
>
> The reviewer's question is how we set $n$ to be sufficiently large in experiments, to ensure it to be unbiased and what would happen if it is not unbiased with $n$ not sufficiently large.
>
> We believe that it is a misunderstanding of our proof in Appendix A.1 and Proposition 1. $n$ in A.1 is a theoretical concept and not a real value usable in experiments. In experiment, we are always using an algorithm with $n = \infty$. Thus, the proposed stochastic are always ensured to be unbiased (either up to all terms when the assumption holds **OR** up to machine precision when the assumption does not hold).
>
> We have revised the proof in the appendix (highlighted in blue) and clarified that `n` is a theorectical concept and in our empirical experiments $n = \infty$.
>
> ---
> > **W3:** The setting of the experiments could be improved to better reflect the contributions. Do the results for all implicit GNNs come from full-batch training on all benchmarks? What would performance be if we incorporated different mini-batching sampling strategies for the baseline implicit GNNs? Meanwhile, more recent baselines of scalable GNN should be added for comparison such as [5-6].
>
> For our experimental results in the submission, USP uses mini-batch training as in their paper and IGNN and MGNNI use full-batch training.
>
> We incorporated different mini-batch sampling methods for the baseline implicit GNNs. The results on Flickr are following:
> | |SEIGNN| IGNN + GraphSAGE | MGNNI + GraphSAGE | IGNN + ClusterGCN | MGNNI + ClusterGCN | IGNN + full batch | MGNNI + full batch |
> |--|--|--|--|--|--|--|--|
> |Accuracy|0.5575|0.5203|0.5219|0.5259|0.5278|0.5302|0.5352|
>
> We can see that baseline implicit GNNs with different mini-batch sampling are slightly worse than using full-batch training, and our method SEIGNN outperforms them.
>
> For more recent baselines of scalable GNNs, We would like to mention again that our goal is not beating all scalable GNNs. Instead, we focus on designing more  scalable, effective, and efficient **implicit** GNNs along with more insights on this topic. We believe that this direction is valuable and our submission may be a valuable step towards scale **implicit** GNNs.

---

> ### Author Response · Authors · 2023-11-21
> **Response to Reviewer K2Aq [2/2]**
>
> [Part 2/2]
> > **W4:** This manuscript aims to scale implicit GNNs to large graphs, but the benchmarks have nearly become standard for even some GNNs that are not specifically designed for scalability. Therefore, large graphs such as ogbn-papers100M should be evaluated for the scalability aim. Note that even the largest dataset (ogbn-products) used in this manuscript is marked as medium in OGB benchmarks.
>
> For choosing datasets in experiments, we mainly follow USP [1] and use ogbn-products as the largest dataset in our experiments. We agree that ogbn-products might be standard for traditional/explicit GNNs. However, it is not easy for implicit GNNs which previously only did full-batch training before USP [1].
>
> ogbn-papers100M is a very large graph with 100M nodes and only a few decoupling-based traditional/explicit GNNs can work on it. Even for explicit GNNs, training efficiency on ogbn-papers100M is also a problem. We agree that the aim to scale up implicit GNNs to handle graphs with 100M nodes is important. However, it requires further improved training efficiency for implicit GNNs, which we will leave for future work.
>
> [1] Unbiased Stochastic Proximal Solver for Graph Neural Networks with Equilibrium States, ICLR 2023.
>
>
> ----
> Thank you again for your valuable feedback. Hope our response answers your questions and we are willing to discuss/clarify further if you have any other questions.

---

> ### Author Response · Authors · 2023-11-22
> **A gentle reminder for the closing Author/Reviewer discussion period**
>
> Dear Reviewer K2Aq,
>
> We would like to thank you again for your helpful review of our work. As the deadline for discussion draws near, we are eager to hear whether our response has addressed your concerns and any other feedback you may have. We are willing to discuss/clarify further if you have any remaining concerns.

---

> > ### Comment · Reviewer_K2Aq · 2023-11-23
> > **Response to the authors**
> >
> > Thank you for your response and the additional discussions.
> >
> > * The comparison with VEQ is important and essential to demonstrate the novelty of SEIGNN. In my opinion, conference proceedings are also not available without purchase is not a reason for not aware of VEQ since most of researchers can access it through their institutions.
> >
> > * Given scalability is the main focus and contribution of this paper, the evaluation on large-scale graphs with billions of nodes (such as ogbn-papers100M) should be taken, otherwise the industrial application of SEIGNN on practical scenarios is likely impossible.
> >
> > * It is still valuable to compare with the SOTA scalable GNNs, since it could help understand the gap between implicit GNNs and regular GNNs on large-scale graphs and motivate further researches.
> >
> > Due to these concerns, I choose to maintain my score.

---

> > > ### Author Response · Authors · 2023-11-23
> > > **Further clarification**
> > >
> > > Dear Reviewer K2Aq,
> > >
> > > Thank you again for your feedback. We would like to make further clarifications.
> > >
> > > > The comparison with VEQ is important and essential to demonstrate the novelty of SEIGNN. In my opinion, conference proceedings are also not available without purchase is not a reason for not aware of VEQ since most of researchers can access it through their institutions.
> > >
> > > We agree that VEQ [1] is relevant and we have already discussed the comparison between VEQ and our method in the previous response. Could you provide any specific feedback on that?
> > >
> > > We summarize the differences between VEQ and our method SEIGNN again for your convenience as follows. 1) VEQ recycles the equilibrium calculated from previous model updates as an initialization for speeding up training while our work proposed a new solver to obtain an unbiased approximation of equilibrium. 2) For mini-batch training, VEQ utilizes the latest equilibrium of in-batch nodes and their 1-hop neighbors with a memory bank for update while our work SEIGNN designs coarse nodes with subgraph sampling minibatch to facilitate global information propagation.
> > >
> > > Note that SEIGNN is also compatible with different sampling methods as shown in Table 6. For accuracy, we observe that SEIGNN is better than VEQ on some datasets such as Flickr, Reddit, Yelp, and ogbn-arxiv (i.e., outperforming VEQ on 4 out of 6 datasets).
> > >
> > > Besides, we would like to clarify that, by saying "Big Data conference proceedings are also not available without purchase", we did not intend to make excuses but just point out the fact. Although you mentioned "most of researchers can access it through their institutions", our institution unfortunately truly does not have access to Big Data conference proceedings. We feel sorry about that. In our humble opinion, we feel that the community is/should be open, diverse and inclusive (thus, ICLR uses openreview) and this kind of barrier is not necessary. Nonetheless, we have overcome all difficulties to get the full text of VEQ paper and respond to your question in the previous post. Thus, we sincerely look forward to your feedback on that.
> > >
> > >
> > >
> > >
> > > [1] Chen, Qi, Yifei Wang, Yisen Wang, Jianlong Chang, Qi Tian, Jiansheng Yang, and Zhouchen Lin. "Efficient and Scalable Implicit Graph Neural Networks with Virtual Equilibrium." Big Data 2022.
> > >
> > > ---
> > > > Given scalability is the main focus and contribution of this paper, the evaluation on large-scale graphs with billions of nodes (such as ogbn-papers100M) should be taken, otherwise the industrial application of SEIGNN on practical scenarios is likely impossible.
> > >
> > > Although we agree that "industrial application of implicit GNNs on practical scenarios" is important, we did not claim that we are aiming for industrial-level scalability. Instead, we think that there is still some way to go to make existing implicit GNNs (including USP and our SEIGNN) scalable to industrial applications. And towards this eventual goal, we believe that our work can be a valuable step and provide some insights to the community.
> > >
> > > ---
> > > > It is still valuable to compare with the SOTA scalable GNNs, since it could help understand the gap between implicit GNNs and regular GNNs on large-scale graphs and motivate further researches.
> > >
> > > In the following, we compare SEIGNN with shadow-GNN [1] as you suggested:
> > > | | Flickr | Reddit | Yelp |
> > > |--|--|--|--|
> > > |SEIGNN|0.5575| 0.9776 | 0.6688 |
> > > |Shadow-GNN|0.5364|0.9713|0.6575|
> > >
> > > We can see that SEIGNN generally outperforms Shadow-GNN on these datasets. For Sketch-GNN, it is from a totally different angle to make GNN more scalable (i.e., using sketch not sampling-base methods). Thus, we do not compare it with SEIGNN here.
> > >
> > > [1] Decoupling the Depth and Scope of Graph Neural Networks, NeurIPS 2021
> > > [2] Sketch-GNN: Scalable Graph Neural Networks with Sublinear Training Complexity, NeurIPS 2022
> > >
> > > ---
> > > Thank you again for your feedback. We hope you can provide more specific feedback, especially for the comparison between SEIGNN and VEQ if that is still your concern. We appreciate your valuable suggestions and we will improve our manuscript accordingly in a later phase (due to the tight time constraints in ICLR this year)

---

### Official Review · Reviewer_4DC8 · 2023-10-30

**Soundness:** 2 fair
**Presentation:** 3 good
**Contribution:** 3 good
**Rating:** 6
**Confidence:** 3

**Summary:**

This paper proposes a scalable and effective implicit GNN (SEIGNN) with a mini-batch training method. Experiments demonstrate SEIGNN outperforms the state-of-the-art implicit GNNs on various large graphs.

**Strengths:**

1. SEIGNN is fast and memory-efficient.
2. The accuracy of SEIGNN on the large datasets is significantly higher than the state-of-the-art implicit GNNs.

**Weaknesses:**

My major concern is the reproducibility of this paper.

1. Table 4 shows that the improvement of SEIGNN is due to the coarse nodes. However, the authors do not analyze why the coarse nodes improve the prediction performance.
2. The authors do not provide the codes for reproducibility.

**Questions:**

See Weaknesses.

---

> ### Author Response · Authors · 2023-11-20
> **Response to Reviewer 4DC8**
>
> We appreciate reviewer 4DC8 for the valuable feedback and suggestions. We would like to clarify the questions as below.
>
> For the reason why the coarse nodes improve the prediction performance, we have provided some analyses with Figure 3 and the related text descriptions/explanations on Page 9. It shows that coarse nodes can improve performance more for low-degree nodes compared with relatively high-degree nodes. Coarse nodes help facilitate long-range information propagation which helps low-degree nodes to receive sufficient information.
>
> We agree that more in-depth analyses from different perspectives would be an interesting direction to explore further. For example, the shortest distances between two nodes have been changed by adding coarse nodes and how it may make differences in terms of performance. However, due to time limitations, we leave it for future work.
>
> For the code implementation, we agree that reproducibility is important for the community and we will definitely make the implementation publicly available upon acceptance of our submission.
>
> We would like to thank you again for your valuable feedback. Hope our response answers your questions and we are willing to discuss/clarify further if you have any other questions.

---

### Official Review · Reviewer_Jpev · 2023-10-31

**Soundness:** 3 good
**Presentation:** 4 excellent
**Contribution:** 3 good
**Rating:** 6
**Confidence:** 3

**Summary:**

The authors identify 2 problems with mini-batching implicit GNNs:

1) The traditional minibatches formed for GNN training is often done through sampling which ignores long-range dependency nodes.

2) Implicit GNNs take a long time to converge, hurting scalability.

The authors propose 2 solutions to said problems.

1) The authors propose adding coarse nodes between minibatch subgraphs to facilitate long-range message propagation during training (when combined with standard techniques like GraphSAGE)

2) The authors extend Neumann series, a implicit GNN solver proposed in [1], by making it a unbiased stochastic solver.

The authors provide experiments and ablations to show their minibatch sampling method is superior to existing works.

[1] Eignn: Efficient infinite-depth graph neural networks

**Strengths:**

The authors provide a compelling argument that their augmentations to mini-batch sampling and implicit GNN solving can improve the scalability of implicit GNNs.

- The authors identified a compelling issue with existing mini-batch approaches.

- The solutions the authors provide are simple, yet intuitive.

- The paper is written clearly and is easy to follow.

- The authors provide timing experiments to empirically verify SEIGNN's scalability.

- The authors try various different minibatch methods with the coarse nodes.

- The method is targeted towards implicit GNNs, which is a subset of all possible GNNs (though the authors demonstrate the method generalizes to other GNNs)

**Weaknesses:**

To be fully convinced, I have a few more questions regarding the approach:

- [important] On datasets where full-batch training is available, what is the performance trade-off of using SEIGNN over the full implicit GNN and a naive subsampling of the full-batch?

- How does modifying the sizes of the minibatch subgraphs affect the efficacy of SEIGNN's coarse nodes?

- How important is using PPR for the coarse node idea? Could SEIGNN generalize to other importance metrics?

- The method is targeted towards implicit GNNs, which is a subset of all possible GNNs (though the authors demonstrate the method generalizes to other GNNs)

**Questions:**

In addition to the weaknesses section, I have a few more questions:

- Have you considered adding multiple coarse nodes in between mini-batch subgraphs? What would be the effect of this?

- What would be the effect of removing/keeping the coarse nodes post-training?

- Why is the stochastic solver slightly better in performance than the Neumann solver in Table 4 of the ablation studies? Is the Neumann solver a truncated version here?

---

> ### Author Response · Authors · 2023-11-20
> **Response to Reviewer Jpev [1/2]**
>
> [Two parts of response: 1/2] We appreciate reviewer Jpev for the valuable feedback and suggestions. We would like to clarify and answer the questions as below.
> > **W1:** [important] On datasets where full-batch training is available, what is the performance trade-off of using SEIGNN over the full implicit GNN and a naive subsampling of the full-batch?
>
> Thanks for the good question. We would like to first point out that the results of baseline implicit GNNs in Table 1 of our submission is using full-batch training.
>
> To answer the question, we conduct the experiments on Flickr dataset to compare SEIGNN with 1) baseline implicit GNNs with full-batch training, 2) baseline implicit GNNs with random sampling of a full batch, 3) baseline implicit GNNs with GraphSAGE (neighbor sampling).
>
> | | SEIGNN | IGNN + full batch | MGNNI + full batch |IGNN + random sampling | MGNNI + random sampling | IGNN + GraphSAGE | MGNNI + GraphSAGE |
> |:---:|:------:|:--:|:--:|:--:|:--:|:--:|:--:|
> |Accuracy|0.5575|0.5302|0.5352|0.4971|0.5008|0.5203|0.5219|
>
> From these results, we can see SEIGNN generally outperforms baseline implicit GNNs with different variants. Comparing full-batch version and mini-batch version of implicit GNNs, full-batch version usually performs better and neighbor sampling version is better than naive random sampling.
>
> Besides, we conduct experiments to show the results of SEIGNN with our designed mini-batch + coarse nodes, SEIGNN with full-batch training, and SEIGNN with sampling methods.
>
> | | SEIGNN (with our mini-batch) | with full-batch | with GraphSAGE sampling | with random sampling |
> |--|--|--|--|--|
> |Accuracy| 0.5575 |0.5392 | 0.5251 | 0.5037 |
>
> We can see a similar trend: full-batch is better than naive sampling methods (i.e., random sampling and GraphSAGE sampling). Additionally, SEIGNN with designed mini-batch coarse nodes outperforms the full-batch version and variants with naive samplings.
>
>
>
> > **W2:** How does modifying the sizes of the minibatch subgraphs affect the efficacy of SEIGNN's coarse nodes?
>
> We conduct the experiments by varying the batch size in a mini-batch subgraph (i.e., the number of target nodes). The results on Flickr are shown below.
>
> | Batch Size | 8192 | 4096 | 2048 | 1024 | 512 | 256 |
> | ---------- | -- | -- | -- | -- | -- | -- |
> | Accuracy   | 0.5564 | 0.5560 | 0.5509 | 0.5425 | 0.5078 | 0.4514 |
>
> We can see that using a very small batch size would lead to bad performance while using the batch size 8192 and 4096 have good accuracy.
> The reason of the bad performance of using a small batch size can be that a small batch size makes the information constrained in a small subgraph, which prohibits the information propagation from distant nodes.
>
>
>
> > **W3:** How important is using PPR for the coarse node idea? Could SEIGNN generalize to other importance metrics?
>
> In our submission, we follow Shadow-GNN [1] to use PPR as an importance metrics to form a subgraph. As pointed out in [1], we can easily extend the approach by using other metrics such as katz index, SimRank [2], and feature similarity, which we leave as future work due to time imitations.
>
> [1] Decoupling the Depth and Scope of Graph Neural Networks. NeurIPS 2021.
>
> [2] Simrank: a measure of structural-context similarity. KDD 2002
>
>
>
> > **Q1:** Have you considered adding multiple coarse nodes in between mini-batch subgraphs? What would be the effect of this?
>
> We are not very sure about the meaning of "in between mini-batch subgraphs". We answer the questions assuming that we are talking about having multiple coarse nodes for each subgraph (multiple coarse nodes represent a subgraph). If not, could you please clarify the question?
>
> We think that this would be an interesting direction to explore for future work. Intuitively, multiple coarse nodes for a single subgraph can facilitate the information propagation within the subgraph when the subgraph is large. However, in another way, multiple coarse nodes for a single subgraph might be similar to a single coarse node for a subgraph but having more subgraphs/partitions with relatively smaller graph sizes doing graph partitioning. If we add more multiple coarse nodes for a subgraph, it is also interesting to explore the design for how to connect coarse nodes within the same subgraph. Due to the time limitation, we haven't conducted the experiments for this and we leave it for future work.
>
>
>
> > **Q2:** What would be the effect of removing/keeping the coarse nodes post-training?
>
> We compare these two during the testing phase.
>
> On Flickr, in testing, accuracy of removing coarse nodes: 0.5575, keep coarse nodes: 0.5551. On Reddit, in testing, removing coarse nodes: 0.9776, keeping coarse nodes: 0.9778.
> We can see that either removing or keeping coarse nodes would have similar performance. Thus, we believe that either way can be fine for inference/test.

---

> ### Author Response · Authors · 2023-11-20
> **Response to Reviewer Jpev [2/2]**
>
> > **Q3:** Why is the stochastic solver slightly better in performance than the Neumann solver in Table 4 of the ablation studies? Is the Neumann solver a truncated version here?
>
> Answer:
> Yes. Here, the Neumann solver is a truncated version of the infinite sum (as shown in Eq(7)). The better performance of the stochastic solver compared with the Neumann solver is caused by the unbiased approximation of the equilibrium obtained using the stochastic solver. We theoretically proved that the stochastic solver provides an unbiased approximation in Proposition 1 of our submission. In contrast, using the Neumann solver would incur some approximation errors of the equilibrium in expectation, which leads to slightly worse performance.
>
>
> Thank you again for your valuable feedback. Hope our response clarifies your questions and we are willing to discuss/clarify further if you have any other questions. Hope you may consider increasing the rating to support our submission if you are satisfied with our response and think our submission is worth publishing.

---

> ### Author Response · Authors · 2023-11-22
> **A gentle reminder for the closing Author/Reviewer discussion period**
>
> Dear Reviewer Jpev,
>
> We would like to thank you again for your helpful review of our work. As the deadline for discussion draws near, we are eager to hear whether our response has addressed your concerns and any other feedback you may have. We are willing to discuss/clarify further if you have any remaining concerns.

---

> ### Comment · Reviewer_Jpev · 2023-11-22
> **Response to Authors**
>
> Thank you for your detailed rebuttal.
>
> I find the method more convincing after "the naive subsampling experiment" (W1) and "the minibatch subgraph size ablation study" (W2). The other answers seem sound as well. Yes, by "multiple in-between nodes", I mean if we expand the many-to-one relationship between subgraph nodes and a single coarse node, either by adding dummy coarse nodes or additional coarse nodes per subgraph.
>
> Conditioned on the inclusion of "the naive subsampling experiment" and "the minibatch subgraph size ablation study", I believe the strengths of the paper outweigh its weaknesses. Thus, I have increased my score.

---

> > ### Author Response · Authors · 2023-11-22
> >
> > Thank you for raising your score! We really appreciate all the helpful suggestions you provided during the discussion phase. We will improve the future version of our manuscript according to your suggestions.

---

### Meta-Review · Area_Chair_t9vh · 2023-12-06

**Metareview:**

The authors propose a new acceleration method for implicit graph neural networks with better empirical results.

Strength:
1. The paper proposes a novel way to scale up implicit GNNs to large graphs while maintaining implicit GNNs long-range dependence.
2. The performance and acceleration rate of their method is satisfying.

Weaknesses:
They focus on linear implicit graph neural networks which makes the problem a little easier and their solution's application less extensive. Also as pointed out by the authors, they cannot use it on ogbnpapers-100M

Decision:
During the discussion, I think most of the concerns are addressed. And 3 of 4 reviewers show a positive attitude for this paper. Therefore, I think this paper can be accepted as an ICLR poster.

**Justification For Why Not Higher Score:**

The scenario is still somehow limited. As pointed out by the authors, they cannot use it on ogbnpapers-100M. How to implement implicit graph neural networks on such a dataset still needs exploring.

**Justification For Why Not Lower Score:**

Their methods are effective and most of the reviewers are willing to accept this paper.

---

### Decision · Program_Chairs · 2024-01-16

Accept (poster)